# Multiple Sclerosis: Enzymatic Cross Site-Specific Recognition and Hydrolysis of H3 Histone by IgGs against H3, H1, H2A, H2B, H4 Histones, Myelin Basic Protein, and DNA

**DOI:** 10.3390/biomedicines10102663

**Published:** 2022-10-21

**Authors:** Georgy A. Nevinsky, Valentina N. Buneva, Pavel S. Dmitrenok

**Affiliations:** 1Institute of Chemical Biology and Fundamental Medicine of the Siberian Division of Russian Academy of Sciences, Lavrentiev Ave. 8, 630090 Novosibirsk, Russia; 2Pacific Institute of Bioorganic Chemistry, Far East Division, Russian Academy of Sciences, 690022 Vladivostok, Russia

**Keywords:** human blood sera antibodies–abzymes, hydrolysis of the H3 histone, IgGs against the H1, H2A, H2B, H3, and H4 histones, enzymatic cross recognition and hydrolysis

## Abstract

Histones have a specific key role in the remodeling of chromatin and gene transcription. In the blood, free histones are damage-connected proteins. Myelin basic protein (MBP) is the major component of the myelin-proteolipid sheath of axons. Antibodies possessing enzymatic activities (abzymes, ABZs) are the specific features of several autoimmune pathologies. IgGs against five histones, MBP, and DNA were obtained from the sera of multiple sclerosis (MS) patients using several affinity chromatographies. The sites of H3 histone splitting by Abs against five individual histones, MBP, and DNA were revealed by MALDI mass spectrometry. It was shown that the number of H3 splitting sites by IgGs against five various histones is different (number of sites): H3 (11), H1 (14), H2A (11), H4 (17), MBP (22), and DNA (29). IgGs against five different histones hydrolyze H3 at different sites, and only a few them coincide. The main reason for the enzymatic cross-reactivity of Abs against H3 and four other histones, as well as MBP, might be the high level of these proteins’ homology. The effective hydrolysis of the H3 histone at 29 sites with IgGs against DNA can be explained by the formation of chimeric abzymes against hybrid antigenic determinants formed by different histones and MBP at the junction of these protein sequences with DNA. The active centers of such abzymes contain structural elements of canonical DNases and proteases. Since free histones are pernicious proteins, antibodies–ABZs against five histones, MBP, and DNA could have a negative role in the pathogenesis of MS and probably other various autoimmune diseases.

## 1. Introduction

Antibodies (Abs) to stable analogs of reaction transition states and natural auto-Abs with catalytic activities are called abzymes (ABZs), and they are successfully described in the literature [1,2,3,4,5,6]. The spontaneous and antigen-stimulated evolution of different autoimmune diseases (AIDs) is associated with the production of abzymes against polysaccharides, lipids, peptides, proteins, DNAs, and RNAs and their complexes. In the blood sera of autoimmune disease patients, a multiplicity of abzymes directly against various antigens was found, mimicking the transition states of chemical reactions. Secondary anti-idiotypic auto-abzymes to the active sites of classical enzymes were also revealed, which may be explained using Jerne’s model of the anti-idiotypic network [7]. The appearance of ABZs in the blood sera is very clear, and the earliest indicator of the autoimmune processes occurs in humans and mammals [1,2,3,4,5,6]. To date, different Abs (IgGs, IgA, and IgMs) hydrolyzing DNAs, RNAs [8,9,10,11,12], poly and oligosaccharides [13,14,15], various peptides, and proteins [16,17,18,19,20,21,22,23] have been discovered in the blood of patients with different AIDs and several viral diseases [1,2,3,4,5,6]. 

Some healthy humans and mammals produce antibody–abzymes with low vasoactive intestinal peptide—[16], thyroglobulin—[18], and polysaccharide-hydrolyzing [13,14,15] activities. However, healthy humans and patients with some pathologies demonstrating insignificant autoimmune reactions usually lack abzymes [1,2,3,4,5,6]. Nonetheless, the germline Abs of some healthy humans could possess amyloid- and superantigen-directed catalytic activities [24,25]. 

MBP is the major protein of the myelin-proteolipid sheath of axons. The specific ABZs against MBP can assault and hydrolyze the MBP of the myelin sheath of axons, possessing an essential negative role in the multiple sclerosis pathogenesis due to the infringement of nerve impulse conduction [1,2,3,4,5,6,21,22]. Histones and their different modified forms hold a vital role in the functioning of chromatin. Free extracellular histones are damaging proteins causing toxic effects through inflammatory pathways and the interaction with Toll-like receptors [26]. ABZs splitting five histones and MBP were found in the blood of HIV-infected [21,22,27,28,29,30,31,32,33,34], SLE [35], and MS [36] patients and in autoimmune mice with experimental encephalomyelitis [37]. In AIDs patients, many anti-histones and anti-DNA antibodies are directed against histone-DNA complexes appearing in the blood due to cell apoptosis [38]. The catalytic cross-reactivity of abzymes against MBP and histones is dangerous to mammals because all histones, due to cell apoptosis, constantly occur in human blood. Considering this, the analysis of the possible catalytic cross-reactivity of antibodies–abzymes against MBP and histones is critical.

The unspecific complexation of some antigens with Abs against different foreign molecules is a widely distributed phenomenon [39,40,41,42]. Specific for different substrates, canonical enzymes usually catalyze only one chemical reaction [43,44,45]. All of the antibodies–abzymes against various proteins described to date usually only cleavage their specific proteins [1,2,3,4,5,6]. The first example of enzymatic cross-reactivity was that of anti-MBP IgGs against five histones (H1, H2A, H2B, H3, and H4) from the sera of HIV-infected patients [32,33,34]. It was proposed that, in the case of HIV-infected patients, the main reason for the hydrolysis of MBP by Abs against histones, and vice versa, is the high level of homology of MBP and histone protein sequences [32,33,34]. In addition, the protein sequences of all five histones are also highly homologous. We have suggested that if abzymes against H1, H2A, H2B, H3, and H4 histones have catalytic cross-reactivity with ABZs against MBP, then they potentially can hydrolyze not only their specific histone but other histones as well. In addition, in the blood, the main antigen stimulating the production of antibodies against DNA is its complexes with histones [38]. The complex of histones between themselves and DNA can potentially provide the formation of new antigenic determinants at the junction of different histones, as well as at the sites of histones’ protein sequence contacts with DNA. The formation of autoantibodies and abzymes to such chimeric antigenic determinants can lead to the production of Abs, the active centers of which could recognize both different histones and DNA and possess proteolytic and DNase activities.

An analysis of the possible complexation polyreactivity and enzymatic cross-reactivity of Abs against the five histones (H1–H4) themselves has been carried out to date only partially. Using MALDI-TOFF mass spectrometry, it was shown that H2A and H2B are hydrolyzed by antibodies against H1, H2A, H2B, H3, H4, MBP, and DNA, and the sites of H2A and H2B histones by these IgGs only partly match [46,47]. 

In this work, an analysis of the ability of the Abs from MS patients against the individual H3, H1, H2A, H2B, and H4 histones, MBP, and DNA to hydrolyze the H3 histone was performed for the first time. 

## 2. Results 

### 2.1. Purification of Antibodies

In this study, we used previously analyzed, electrophoretically homogeneous IgG preparations from the sera of fifteen multiple sclerosis patients, which were isolated by sequential chromatography of the blood plasma proteins—first, on Protein G-Sepharose, using conditions allowing for the removal of nonspecifically bound proteins [27,28,29,30,31,32,33,34]. Then, polyclonal IgG preparations were subjected to FPLC gel filtration under drastic conditions (pH 2.6), destroying immune complexes according to [27,28,29,30,32]. To analyze the “average” site-specific splitting of H3 by IgGs against the H1–H4 histones, we have obtained a mixture of equal amounts of fifteen IgG preparations (IgG_mix_) possessing high activities in the splitting of histones, MBP, and DNA. Any possible artifacts due to potentially possible traces of contaminating classical proteases were excluded earlier [31]; the IgG_mix_ preparation was separated by SDS-PAGE, and its proteolytic and DNase activities in the hydrolysis of histones, MBP, and DNA were found in only one IgG_mix_ protein band. 

IgGs against MBP were previously isolated from IgG_mix_ by affinity chromatography on MBP-Sepharose. The IgG fraction nonspecifically bound to the affinity sorbent was first eluted with 0.2 M NaCl. Specific anti-MBP IgGs with a high affinity for MBP were eluted using 3.0 M NaCl and Tris-Gly buffer, pH 2.6 [46,47]. For the additional purification of anti-MBP IgGs from the potential impurities of the IgGs against five histones, the anti-MBP IgG fraction was passed through histone5-Sepharose (immobilized five histones). The fraction obtained at the loading onto histone5-Sepharose was further used as the anti-MBP IgGs. 

The fractions of IgGs eluted from MBP-Sepharose on the loading were combined and passed twice through MBP-Sepharose and then used to obtain IgG fractions against five histones using histone5-Sepharose. Then, the IgGs against five histones were applied sequentially to five sorbents with five immobilized individual histones: H3, H1, H2A, H2B, H3, and H4. The fraction eluted upon loading, without affinity for the previous sorbent, was used for each of the mentioned subsequent chromatographies. Finally, we prepared five IgG preparations against five individual histones: anti-H3, anti-H1, anti-H2A, anti-H2B, and anti-H4 IgGs [46,47].

### 2.2. SDS-PAGE Analysis of Histones and MBP Hydrolysis 

Polyclonal IgGs from the blood of HIV-infected patients, as shown earlier [27,28,29,30,32], and patients with multiple sclerosis effectively split the five human histones [32] and MBP [21,22]. Moreover, the IgGs of HIV-infected patients against MBP and five histones demonstrated not only polyspecific complexation but also catalytic cross-reactively in the hydrolysis of five histones and MBP [28,29,30]. It was interesting whether the IgGs of multiple sclerosis patients against histones are also capable of splitting the five histones and MBP and vice versa. Moreover, it seemed important to check whether it is possible to form chimeric antibodies against histone complexes with DNA, the active centers of which are capable of hydrolyzing both histones and DNA.

To analyze this kind of enzymatic cross-reactivity, we previously used the fraction of anti-histone IgGs (eluted from histone5-Sepharose), anti-MBP IgGs (eluted from MBP-Sepharose), and anti-DNA IgGs (eluted from DNA-cellulose). Appendix A demonstrates the hydrolysis of five histones by anti-histones, anti-MBP, and anti-DNA IgGs, while Appendix A shows the hydrolysis of MBP by these IgGs. 

These data may potentially favor the idea that the anti-MBP, anti-histones, and anti-DNA IgGs of MS patients could possess a known phenomenon of unspecific complex formation due to their polyreactivity [39,40,41,42] and mutual enzymatic cross-reactivity in MBP and histones hydrolysis. These results, however, cannot provide absolute evidence of enzymatic cross-reactivity between IgG–abzymes against five histones, MBP, and DNA because, even after their purifications using several affinity chromatographies, it cannot be excluded that isolated Abs nevertheless could contain very small admixtures of alternative IgGs. The best evidence of enzymatic cross-reactivity may be the undeniable difference in the specific sites of the histones hydrolysis by IgGs against MBP and against these five histones. However, the principal objective of this study was not only to reveal the possibility of the cross-hydrolysis of histones, MBP, and DNA by IgGs against these proteins and DNA but, first of all, to find out, using MS patients’ Abs, whether there is an enzymatic cross-activity of Abs against each of the individuals to the four other histones, MBP, and DNA. In this study, we first checked the possibility of the hydrolysis of the histone H3 with specific IgG–abzymes against the H1–H4 histones, MBP, and DNA. 

As shown previously [46,47], not only Abs against histones and MBP but also IgGs against DNA are able to effectively hydrolyze histones and MBP (Appendix A). It was very interesting to ask whether IgGs that have a high affinity for DNA, MBP, and histones can hydrolyze DNA. The analysis of DNA hydrolysis with all IgGs was performed. As an example, Appendix A shows the data of the analysis of scDNA hydrolysis by several IgGs. One can see that all IgGs hydrolyze DNA [46,47].

These data indicated that, as in the case of SLE monoclonal antibodies against MBP obtained by Phage display [48,49,50,51,52,53,54,55], the active centers of some monoclonal antibodies in the fractions of polyclonal MS patients’ IgGs against histones, MBP, and DNA could combine at least two activities—protease and DNase.

The data of Appendix A may potentially demonstrate that the anti-MBP, anti-histones, and anti-DNA IgGs of MS patients could possess a known phenomenon of complex formation polyreactivity, as revealed earlier [39,40,41,42], and they recognize all these molecules and hydrolyze them. As mentioned above, these findings, however, cannot provide truthful evidence of enzymatic cross-reactivity between the IgG–abzymes of MS patients against five histones, MBP, and DNA. Therefore, for more convincing evidence of the existence of this phenomenon, an analysis was conducted of the sites of H3 histone hydrolysis by IgGs against individual histones, MBP, and DNA.

### 2.3. MALDI Analysis of H3 Histone Hydrolysis 

In this work, for the first time, an analysis of the hydrolysis of the H3 histone by antibodies against various histones, MBP, and DNA was carried out. The IgG antibody fractions with a high affinity to five histones, MBP, and DNA were used to reveal the cleavage sites of H3 by MALDI TOFF mass spectrometry. Right after the addition of the IgGs (Figure 1A), the H3 histone was almost homogeneous: only two signals corresponding to one- (*m*/*z* = 15,263.4 Da) and two-charged ions (*m*/*z* = 7631.7 Da) were revealed. 

H3 cleavage assays were carried out with IgGs against all histones after 3–24 h of incubation. Some of the pronounced peaks corresponding to different sites of H3 histone hydrolysis by IgGs against H3 were visible after 6 h of hydrolysis (Figure 1B). The incubation of mixtures for 24 h results in the nearly complete hydrolysis of initial H3 to the same and additional short fragments of the protein (Figure 1C). Based on the analysis of peaks in the 8-9-spectra corresponding to the different times of incubation, eleven sites of the hydrolysis were identified, three of which are major (P38-H39, P43-G44, and Y54-Q55), three of which are moderate (R49-T50, L100-V101, and L100-V101), and five of which are minor splitting sites (Q5-T6, L61-I62, Q68-R69, G102-L103, and F104-E105).

One of the fascinating questions was whether IgGs against the H1, H2A, H2B, and H4 histones could hydrolyze the H3 histone. An unexpected result was obtained after the incubation of H3 with IgGs against the histone H1. Figure 1D demonstrates that after 6 h of incubation of H3 with antibodies against H1, there is the formation of several products of H3 histone hydrolysis by anti-H1 IgGs. As in the case of the hydrolysis of H3 with anti-H3 antibodies, the H3 histone was almost wholly hydrolyzed by anti-H1 IgGs after 24 h of incubation. As a result of the analysis of a large number of spectra, 14 sites of H3 hydrolysis by antibodies against H1 were identified: 4 major ones (P43-G44, R63-K64, S87-A88, and E94-A95), 7 average ones (E50-I51, P66-F67, F67-Q68, E73-I74, I74-A75, H113-F114, and I130-R131), and 3 minor ones (Q5-T6, G12-G13, and K56-S57). 

Figure 1E shows the MALDI spectra of the products after the hydrolysis of the H3 histone with anti-H2A IgGs for 10 h. The rate of H3 hydrolysis by Abs against H2A was comparable with that for IgGs against the H3 histone. After 24 h of incubation, a nearly complete hydrolysis of the histone H3 by anti-H2A IgGs was observed (Figure 1E). Based on the analysis of 17 spectra, 11 splitting sites were detected: 3 major sites (P43-G44, Y54-Q55, and L100-V101), 4 moderate sites (R53-R54, Q68-R69, R69-L70, and Q76-D77), and 4 minor sites (Q5-T6, R52-R53, K56-S57, and S87-A88).

The H3 histone was degraded by anti-H2B IgGs approximately 1.5 times slower than it was by anti-H3 antibodies. Still, after 24 h of incubation, it was nearly wholly hydrolyzed to many products, including several short peptides (Figure 2A).

Thirteen reliably detectable cleavage sites were found, among which two were major (P43-G44 and L100-V101), four were average (A31-T32, Y54-Q55, Q55-K56, and Q68-R69), and seven were minor sites (R69-L70, D81-L82, F84-Q85, S87-A88, G102-L103, F104-E105, and I130-R131) (Figure 2A).

Antibodies against the histone H4 also cleaved H3, but at a rate approximately 1.7-fold slower than that of anti-H3 abzymes (Figure 2B). After 24 h of the incubation, all detectable products of H3 hydrolysis, which appeared during incubation from 3 to 10 h, were detected (Figure 2B). Two of seventeen sites were major (P43-G44 and P66-F67), three were average (L60-L61, Q68-R69, and R72-E73), and twelve were minor (Q5-T6, T6-A7, T22-K23, V46-A47, A47-L48, L48-R49, Y54-Q55, K56-S57, Y99-L100, F104-E105, M120-P121, and I130-R131). 

It was interesting to compare the sites of the H3 histone hydrolysis by IgGs against histones and MBP. Figure 3A demonstrates six products of H3 histone hydrolysis by IgGs against MBP after 3 h of incubation. After 24 h of the incubation (Figure 3B), Abs against MBP almost completely hydrolyzed the histone H3. A total of 22 sites of H3 hydrolysis by antibodies against MBP were found: 3 major, 9 average, and 10 minor ones. 

Assuming that hybrid antigenic determinants can be formed at the junction of histone and DNA sequences and that crossbred antibodies against proteins and DNA can be developed against them, we analyzed the hydrolysis of the histone H3 by antibodies with a high affinity for DNA. Figure 3C shows the products of H3 hydrolysis by anti-DNA IgGs for 3 h of incubation, and Figure 3D shows the same for 24 h of incubation. After 3 h of incubation, the formation of only four hydrolysis products is visible, while after 24 h, there are many of them. Based on the totality of the peaks of H3 hydrolysis products from 3 to 24 h, 29 statistically significant sites of its hydrolysis by anti-DNA antibodies were found: 6 major, 7 average, and 16 minor ones. All sites of H3 histone hydrolysis by antibodies against histones, MBP, and DNA are shown in Figure 4. Major cleavage sites are indicated by big stars (★), moderate sites are indicated by arrows (↓), and minor ones are splitted by colons (:).

To simplify the analysis of the overlapping and different cleavage sites of H3 with five IgGs against different histones, MBP, and DNA, all data are collected in Table 1.

Table 1 and Figure 4 show that the H3 hydrolysis sites by anti-H3 histone IgGs are mainly localized in two extended clusters (P43-A75 and V89-E106). 

There are no sites of H3 hydrolysis with anti-MBP antibodies from K37-I51 AAs (Table 1, Figure 4F). In the case of the remaining abzymes against the five histones, the clusters of hydrolysis sites overlap, but the splitting sites are mostly still different (Table 1, Figure 4).

In the hydrolysis of H3 by antibodies against H3, there is a coincidence of different cleavage sites corresponding to IgGs against other histones: anti-H1–3, anti-H2A–5, anti-H2B–6, anti-H4 histone–6, and anti-MBP-3 sites (Table 1). However, the other sites are not the same. All the overlapping H3 hydrolysis sites differ in the case of antibodies against different histones in terms of intensity (major, average, or minor) (Table 1, Figure 4). The only common major site of histone H3 hydrolysis by all antibodies against the five histones is the site P43-G44. Also common to the antibodies against four of the five histones are the sites Y54-Q55 and Q68-R69 (Table 1). This indicates that the IgGs against the histone H3 do not contain at least palpable impurities of abzymes against the four other histones or anti-MBP, and vice versa. Therefore, as in the case of anti-H3 and anti-MBP antibodies from the blood of HIV-infected patients [30], such abzymes of MS patients against histones and MBP possess both the polyreactivity of complex formation and enzymatic cross-reactivity.

Of particular interest is the ability of IgGs with a high affinity for DNA to hydrolyze histones and MBP. As one can see from Table 1 and Figure 4, anti-DNA antibodies hydrolyze the H3 histone at 29 sites. Only 5 out of the 29 sites match those for anti-H4 antibodies. At the same time, out of 29, 14 hydrolysis sites are absent in the case of antibodies against five histones. Approximately the same situation is observed for IgGs against MBP and DNA; there is no coincidence of 16 hydrolysis sites among between 22 and 29 sites for anti-MBP and anti-DNA antibodies. 

## 3. Discussion

The polyreactivity of Abs is a widespread phenomenon [39,40,41,42]. Even some different molecules that are structurally similar to specific antigens can form complexes with antibodies against other specific compounds. This is why, during affinity chromatography, antibodies against a cognate antigen and Abs to some compounds structurally related to this antigen can form complexes with the immobilized antigens [39,40,41,42]. The affinity of Abs for unspecific molecules at the same time is usually significantly lower than that for cognate antigens. Therefore, Abs against non-specific antigens may usually be eluted at affinity chromatography using NaCl at a concentration of 0.1–0.15 M [1,2,3,4,5,6,43,44,45]. Taking this into account, during the isolation of IgGs against five individual histones (H1–H4), MBP, and DNA, we eluted IgGs nonspecifically bound to six immobilized proteins and DNA using 0.2 M NaCl. For the additional purification of the IgGs against five histones, MBP, and DNA, they were additionally passed twice through alternative affinity sorbents. Finally, specific antibody fractions against MBP, five individual histones (H1–H4), and DNA were obtained.

As shown previously [31,46,47], the IgG preparations of MS patients used by us in this work do not contain any canonical proteases or DNases. Moreover, the same incarceration can be drawn using the comparison of histone H3 hydrolysis sites with IgGs against MBP, five histones, and DNA. Trypsin hydrolyzes different proteins after the lysine (K) and arginine (R) residues. The H3 sequence contains 13 Lys and 17 Arg residues presenting 30 potential sites for this histone cleavage by trypsin. However, the number of sites of H3 splitting by all IgG preparations used after Lys and Arg varies mainly from 1 (anti-H3, anti-H2B, and anti-H4 IgGs) to 2 (anti-H1 IgGs and anti-H2A IgGs) (Figure 4, Table 1). Only IgGs against MBP cleavage H3 at 10 sites after Lys and Arg disposed within relatively short and specific clusters (Figure 4, Table 1). Chymotrypsin splits proteins after aromatic amino acids (F, Y, and W). There are seven potential sites for hydrolysis in the H3 histone by chymotrypsin. Anti-H2A IgGs did not demonstrate sites of cleavage after F residue (Table 1). Only one site of splitting after F was found for anti-H3 and anti-H1 (Table 1) IgGs, while anti-H2B and anti-H4 IgGs demonstrated two sites of cleavage after F (Table 1). No hydrolysis sites were found after Y in the case of anti-H1, anti-H2B, and anti-MBP IgGs, while one hydrolysis site was revealed for anti-H3, anti-H2A, and anti-H4 Abs. The cleavage sites of H3 by five IgGs occur mainly in clusters containing neutral non-charged and nonaromatic amino acids (Q, P, L, V, G, I, S, and H) or acidic acids—E and D (Table 1, Figure 4). Thus, overall, the sites of specific hydrolysis of H3 by five IgGs against five histones do not correspond to those for trypsin or chymotrypsin. They are not distributed along the entire length of the protein molecule but are located in specific amino acid clusters.

Interestingly, using a large number of monoclonal Abs of SLE patients, it was shown that their active centers could correspond to thiol, serine, or metal-dependent proteinases [51,52,53,54,55]. However, in contrast to canonical proteases, abzymes hydrolyze specific proteins, mainly in their amino acid clusters corresponding to antigenic determinants [1,2,3,4,5,6,27,28,29,30,31,32,33,34,51,52,53,54,55]. In addition, the cleavage of H3 by five IgG preparations occurs in clusters containing hydrolysis sites not after charged K, R, or aromatic F and Y but mainly after neutral or acidic amino acids (Figure 4, Table 1).

One intriguing result is that MS IgGs not only against the histone H3 but also against the H1, H2A, H2B, H3, and H4 histones and MBP can hydrolyze the histone H3. The primary evidence that the preparations of each IgG to five individual histones and myelin basic protein do not contain at least noticeable impurities of Abs against any of the alternative histones or MBP is that the H3 cleavage sites for each of them are significantly different (Figure 4, Table 1). As was previously shown using antibodies against each of the five histones and MBP from the blood of HIV-infected patients, the main reason for enzymatic cross-reactivity could be provided by the high homology of the protein sequences MBP and histones [28,29,30,31,32,33,34]. Therefore, it was interesting to analyze the general homology between the protein sequence of H3 with four other histones and MBP. 

The complete identity of amino acids between H3 and H1 (three alignments) was from 26.8 to 29.8% (average value 28.3 ± 2.1%), while the similarity (identical together with non-identical amino acids but with highly similar physicochemical properties) varied from 52.3 to 54.3% (average 53.3 ± 1.4%). The identity between H3 and H2A according to three different alignments changes from 27.2 to 27.7% (average value 27.5 ± 0.3%), and similarity changes from 51.9 to 59.4% (average 54.6 ± 4.2%). For the H3 and H2B histones, the following homology values were found: identity—22.6–30.4% (average value 26.5 ± 3.9%); similarity—50.0–53.4% (average value 51.3 ± 1.9.0%). Approximately the same homology data were revealed for the H3 and H4 histones: identity—29.0–31.9% (average 30.3 ± 1.5%); similarity—48.9–51.4% (average value 50.0 ± 1.4%). 

An analysis of the homology of the complete sequences of MBP and five histones was carried out: H3 (identity—22.8–25.3% (average 24.4 ± 1.4%), similarity—44.3–47.6% (average 45.5 ± 1.8%)); H1 (identity—25.4–28.4% (average 26.9 ± 2.1%), similarity—48.8–52.8% (average 50.8 ± 2.8%)); H2A (identity—25.0–26.8% (average 25.9 ± 1.3%), similarity—47.6–50.3% (average 49.0 ± 1.9%)); H2B (identity—24.4–25.4% (average 25.4 ± 2.2%); similarity—46.0–53.2% (average value 49.1 ± 3.5%)); H4 (identity—25.0–29.4% (average 27.2 ± 3.1), similarity—46.2–48.6 (average 47.4 ± 1.1%)). The indices of the identity of the protein sequence of the five histones and MBP (24.4–27%) and the similarity (45.5–52.4%) are also very similar. This may be the reason for the possibility of H3 hydrolysis by IgGs against MBP and five histones (Figure 4, Table 1). 

Most likely, not the general level of homology between the complete sequences of proteins but rather the homology between the sequences, which are hydrolyzed by abzymes, is more important for the manifestation of abzyme catalytic cross-reactivity. An analysis of the homology of the main H3 clusters in which there are the main sites of H3 sequence hydrolysis by antibodies against four different histones with the sequences of these histones (H1, H2A, H2B, and H4) leads to more significant values of identity and similarity (%): H1 (62.5 and 87.5), H2A (50.0 and 83.3), H2B (50.0 and 87.5), and H4 (50.0 and 75.0). Somewhat less identity and similarity are observed between the sequence cleaved in H3 by anti-MBP IgGs and one of the fragments of myelin basic protein: 45.5% identity and 63.6% similarity.

The histones and MBP contain many positively charged residues of lysine and arginine. Such amino acid residues are necessary for histones for the interaction with negatively charged internucleoside groups of DNAs. In addition, it was demonstrated that MBP can also efficiently form complexes with DNAs [56]. Thus, it is possible that many positively charged amino acids in the five histones and MBP can also make an important contribution to the ability of Abs against these proteins to form complexes with foreign histones. Due to the high homology of MBP and the five histone sequences and the interaction with positively charged amino acids, Abs against MBP can form complexes with H3 and other histones and hydrolyze them, demonstrating enzymatic cross-reactivity. 

A special question is why IgGs with a high affinity for DNA effectively hydrolyze not only DNA but also the H3 histone and MBP, as well as why, in the case of anti-DNA antibodies, there are so many sites of H3 histone hydrolysis. In the complex of the five histones and DNA that appear in the blood as a result of cell apoptosis, DNA is in contact with all five histones. In addition, these DNA–histones complexes can partially dissociate in the blood. This may lead to the formation of a wide variety of histone–DNA contacts and the formation of Abs against DNA in complexes with individual histones or a combination of them. As a result of the simultaneous production of abzymes against sequences of different histones and DNA, antibodies with a high affinity for DNA can be gained, the active centers of which contain all the necessary structural elements of canonical DNases and proteases. As can be seen from Table 1, the sites of H3 hydrolysis by anti-DNA antibodies weakly match with those for antibodies against each of the individual histones, but they are in good agreement with the total set of sites of H3 cleavage by antibodies against all five histones and MBP. The large number of sites of H3 hydrolysis by anti-DNA antibodies may be due to the fact that these polyclonal antibodies consist of a set of anti-DNA monoclonal antibodies corresponding to Abs against DNA in its complexes with five different histones and MBP. The presence of 5 of the 29 sites of H3 hydrolysis by anti-DNA antibodies that are not detected for antibodies against the five histones and MBP may be due to the fact that some histones or MBP protein sequences can be changed markedly in complexes with DNA compared to those for DNA-free histones.

## 4. Materials and Methods

### 4.1. Chemicals, Donors, and Patients

All of the chemicals used, including the BrCN-activated Sepharose, the five electrophoretically homogeneous human histones (H1 (M2501S), H2A (M2502S), H2B (M2508S) H3 (M2503S), and H4 (M2504S)), and the equimolar mixture of the histones, were from Sigma (H9250; St. Louis, MO, USA). The sorbent columns ((Superdex 200 HR 10/30 (17-5175-01) and Protein G-Sepharose (17061801)) were obtained from GE Healthcare (GE Healthcare, New York, USA). The human myelin basic protein was obtained from the DBRC Center of Molecular Diagnostics and Therapy (DBRC-HMBP; Russia, Moscow). The Protein Sepharose columns with immobilized histones and MBP were prepared according to the manufacturer’s protocol using BrCN-activated Sepharose (17098101, GE Healthcare), five individual histones or their mixture, and MBP.

Proof that the IgGs of MS patients split five histones and MBP was published earlier using IgG preparations from the blood of 59 patients [21,22,31]. Patient medical characteristics are shown in Appendix A. The diagnosis of MS was established by specialists from the Novosibirsk Medical University Multiple Sclerosis Center based on the classification of McDonald [57]. The disease severity of all 59 multiple sclerosis patients was scored based on Kurtzke’s Expanded Disability Status Scale (EDSS) [58]. The MS patients at entry had no symptoms of any infections. None of the MS patients at the sample collection time had received any anti-disease therapies during the six months before the study. 

The protocol of blood sampling was confirmed by the human local ethics committee (State Medical University at Novosibirsk, Russian federation; number 105-HIV; 07. 2010). This ethics committee supported our study based on the Helsinki ethics committee guidelines. All MS patients provided written agreement to give blood for our scientific purposes. 

In this work, we used a mixture of 15 of the 59 IgG preparations described earlier [31], which demonstrated high activity in the hydrolysis of five histones, MBP, and DNA.

### 4.2. Antibody Purification

Electrophoretically homogeneous preparations of Abs were picked out from the blood plasma of MS patients—first by affinity chromatography of the plasma proteins on Protein G-Sepharose and then using gel filtration in drastic conditions on a Superdex 200 HR 10/30 column, as in [31,32,33,34,46,47]. To protect IgGs from possible contaminations, they were passed through a Millex filter (pore size 0.1 μm). After 6–8 days of storage at 4 °C for refolding, the IgG preparations were used in different assays. The SDS-PAGE analysis of IgGs for homogeneity was performed using 3–17% gradient gels (0.1% SDS); all proteins were visualized by silver staining [31,32,33,34,46,47].

Earlier, after the SDS-PAGE analysis of antibodies, 3–4-mm cross-sections of the gel longitudinal slices were used to obtain eluates, and it was shown that the IgGs obtained from the MS patients were not contaminated with any canonical proteases [31]. Proteolytic activity was revealed only in the eluates corresponding to the gel fragments containing IgGs. 

### 4.3. Affinity Chromatography of IgGs 

To obtain IgGs against individual histones, a mixture of 15 Abs (IgG_mix_) (used in [31,46,47]) exhibiting high activity in the cleavage of the five histones, MBP, and DNA was used. The removal of IgGs against all five histones from a homogeneous IgG_mix_ preparation with no admixture of any classical proteases was reached using MBP-Sepharose (immobilized MBP), equilibrated in 20 mM Tris-HCl, pH 7.5 (buffer A). To obtain the IgGs fraction against MBP, the column was first washed with buffer A to zero optical density (A_280_). Then, the adsorbed IgGs with a low affinity to MBP were eluted from the column using buffer A supplemented with 0.2 M NaCl. Finally, the IgGs with a high affinity for MBP were eluted specifically—first by 3.0 M NaCl and then by 0.1 M glycine-HCl, pH 2.6. The fractions eluted with 3.0 M and acid buffer were subjected to additional purification from potential admixtures of Abs against the five histones. These fractions were combined and were twice put on the column of histone5-Sepharose (immobilized mixture of five histones). The fraction of antibodies eluted from the column upon the loading was designated and used as the anti-MBP IgGs. 

The fraction containing a mixture of IgGs against the five histones, which was eluted from MBP-Sepharose at loading, was subjected to being passed through this sorbent twice and was then used to isolate IgGs against the five individual histones [46,47]. This fraction was applied first on H3-Sepharose containing immobilized H3 histone. The fractions eluted at loading in the case of each of the used sorbents were applied sequentially to the next one: H1 (H1-Sepharose), H2A (H2A-Sepharose), H2B (H2B-Sepharose), and H4 (H4-Sepharose). All chromatographies were carried out as in the case of MBP-Sepharose. IgGs against the H1–H4 histones were specifically eluted from each affinity sorbent with buffer, pH 2.6. These fractions were designated, respectively, as anti-H3, anti-H1, anti-H2A, anti-H2B, and anti-H4 IgGs. 

### 4.4. Affinity Chromatography of IgGs on DNA-Cellulose

To obtain anti-DNA IgGs, the fractions with no affinity for MBP-Sepharose and His-5-Sepharose (eluted from these sorbents upon loading) were also additionally passed twice through these two affinity sorbents. The final fraction after the removal from IgG_mix_ Abs against histones and MBP was used to obtain the anti-DNA IgGs of the MS patients. It was applied on DNA-cellulose (5 mL, equilibrated with 20 mM Tris-HCl buffer, pH 7.5). After the elution of the Abs with a low affinity for DNA 0.2 M NaCl, anti-DNA IgGs were eluted—first with 3.0 M NaCl and then with the acidic buffer. By combining these two factions, anti-DNA IgGs were obtained [46,47].

### 4.5. Proteolytic Activity Assay

The reaction mixtures (9–15 μL) contain 25 mM Tris-HCl (pH 7.5), 0.7–1.0 mg/mL H3 histone or 1.0 mg/mL myelin basic protein, and 0.01–0.1 mg/mL IgGs against one of the five individual histones (H1–H4) or MBP. All mixtures were incubated for 2–24 h at 37 °C. The efficiency of the histone H3 (or all five histones) and MBP cleavage was analyzed by SDS-PAGE using 15% gels in the absence of DTT and under nonreducing conditions, as in [27,28,29,30,31,32,33,34,35,46,47]. The products of H3 and MBP splitting were revealed using gels stained with Coomassie Blue. The gels, after staining, were scanned and quantified, as in [31], using Image Quant v5.2 software. The efficiency of H3 hydrolysis was assessed by reducing the protein content compared to the control-incubation of H3 or MBP in the absence of antibodies. 

### 4.6. DNA Hydrolysis Assay

The reaction mixtures (10–18 μL) contained: 4.0 mM MgCl_2_, 0.2 mM CaCl_2_, 20 mM Tris-HCl, pH 7.5, 10 μg/mL *pBluescript* supercoiled (sc)DNA plasmid, and 2.0–30.0 μg/mL different IgGs, similar to [11,12,46,47]. After incubation for 1–4 h at 37 °C, 2.6 μL of the loading buffer containing 1% SDS, 50 mM EDTA, pH 8.0, 30% glycerol, and 0.005% bromophenol blue was added to the reaction mixture. Electrophoresis was performed using 0.8% gel of agarose until the bromophenol blue migrated in 2/3 of the gel. The scDNA in the gel was stained with ethidium bromide (0.5 μg/mL, 1–2.5 min). The gels were imaged using the Gel Doc gel documentation system Bio-Rad (Bio-Rad, Berkeley, CA, USA). The photographs of the gels were counted using the ImageQuant 5.2 program. The level of DNA-hydrolyzing activity of the IgGs was determined by the degree of hydrolysis of the plasmid scDNA form to a relaxed form of DNA. 

### 4.7. MALDI-TOF Analysis of Abs-Dependent H1 Histone Hydrolysis

H3 hydrolysis by Abs against the five histones (H1–H4), MBP, and DNA was performed using the Reflex III system (Bruker Company; Frankfurt, Germany): a 337 nm VSL-337 ND nitrogen laser with a 3 ns pulse duration. Mixtures (10–12 µL) containing 20 mM of Tris-HCl (pH 7.5), 0.7–0.8 mg/mL of one of the five histones, and 0.04–0.05 mg/mL of one of the IgGs were incubated at 30 °C for 0–24 h. To 1.3 µL of the sinapinic acid matrix mixed with 1.3 µL of 0.2% trifluoroacetic acid, 1.3 µL of the solutions containing the H3 histone before or after incubation with different IgGs against the five histones, MBP, and DNA was added; 1.1–1.2 µL of these mixtures, after loading on the MALDI plates, were air-dried. All of the MALDI mass spectra were calibrated using the standard Bruker Daltonic protein mixtures II and I (Germany) in two modes of calibration: internal or external. The analysis of peptide molecular masses corresponding to specific sites of H3 splitting by IgGs against five histones and MBP was performed using Protein Calculator v3.3 (Scripps Research Institute; La Jolla, CA, USA).

### 4.8. Analysis of Sequence Homology 

The analysis of the protein sequences homology between the histones and MBP was performed using *lalign* (http://www.ch.embnet.org/software/LALIGN_form.html) (accessed on 1 January 2008). 

### 4.9. Statistical Analysis 

The results are presented using the mean ± S.D. of 7–10 independent MALDI mass specters for each sample of H3 hydrolysis by IgGs against five histones, MBP, and DNA (accessed on 1 January 2008).

## 5. Conclusions

Here, we have shown for the first time, using IgG–abzymes from patients with multiple sclerosis, that IgGs against H3, H1, H2A, H2B, H4, human MBP, and DNA possess an ability similar to anti-H3 IgGs to form complexes with the H3 histone, demonstrating polyreactivity in complexation. Moreover, an exciting result was obtained. IgG–abzymes against H1, H2A, H2B, H4, and MBP, and DNA possess catalytic cross-reactivity with anti-H3 antibodies, and all of them are capable of hydrolyzing the histone H3. Evidence that the ability of IgGs against H1, H2A, H2B, H4, MBP, and DNA to hydrolyze the H3 histone is their own property follows from the fact that the sites of hydrolysis of the H3 histone by different IgGs are individual for each preparation of Abs and differ in their location in the protein sequence of the H3 molecule. The main reason for the formation of such polyfunctional abzymes is most likely the existence of DNA–histones complexes, chimeric antigenic determinants consisting of protein and nucleic sequences, and the high level of homology of all five histones between themselves and MBP. Since the histones H1–H4 and their complexes with DNA constantly occur in human blood due to cell apoptosis, the catalytic cross-reactivity of IgGs–abzymes against histones, MBP, and DNA can play a very negative role in MS pathogenesis. 

## Figures and Tables

**Figure 1 biomedicines-10-02663-f001:**
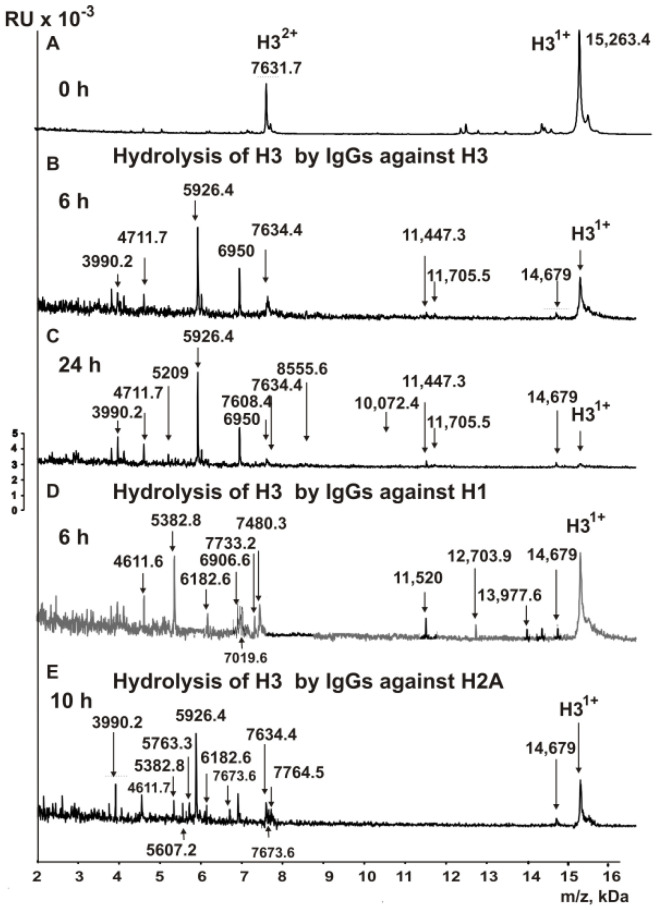
MALDI mass spectra corresponding to the H3 histone (0.8 mg/mL) over the time of hydrolysis (0–24 h) in the presence of different IgGs (0.045 mg/mL) against H3 (**A**–**C**), H1 (**D**), and H2A (**E**). All of the used antibody preparations and the molecular masses (m/z, Da) of the products are indicated on Panels A–E; RU—relative units.

**Figure 2 biomedicines-10-02663-f002:**
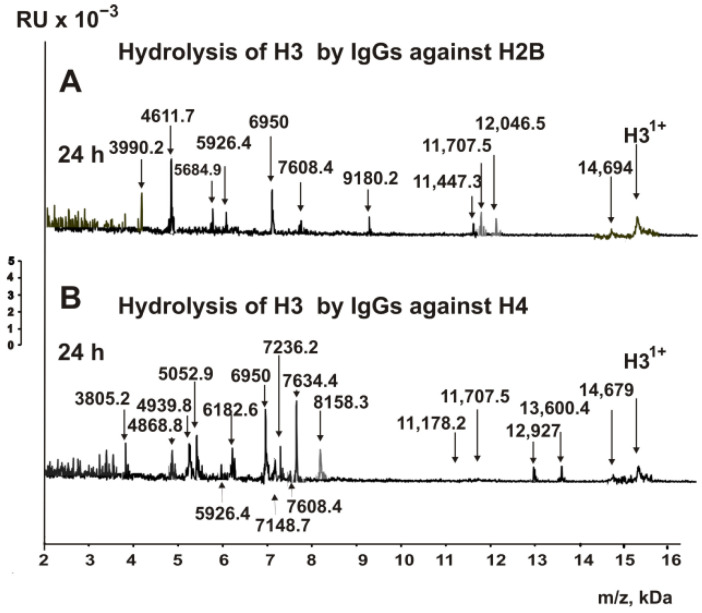
MALDI spectra corresponding to the H3 histone (0.8 mg/mL) over the time of hydrolysis in the presence of IgGs (0.04 mg/mL) against H2B (**A**) and H4 (**B**). All of the used antibody preparations and the molecular masses (*m*/*z*, Da) of the products are indicated on the panels; RU—relative units.

**Figure 3 biomedicines-10-02663-f003:**
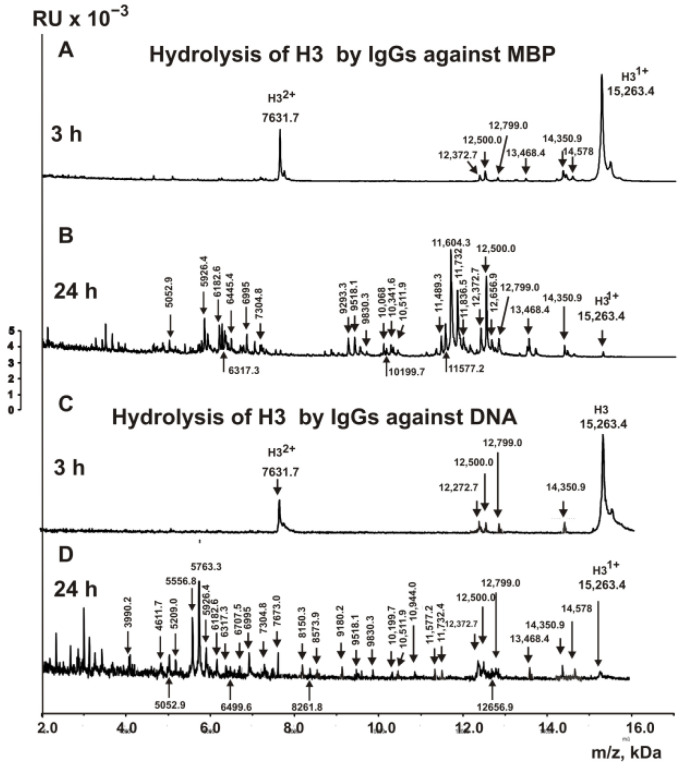
MALDI spectra corresponding to the H3 histone (0.8 mg/mL) over the time of hydrolysis in the presence of IgGs (0.04 mg/mL) against MBP (**A**,**B**) and DNA (**C**,**D**). All of the used antibody preparations and the molecular masses (*m*/*z*, Da) of the products are indicated on the panels; RU—relative units.

**Figure 4 biomedicines-10-02663-f004:**
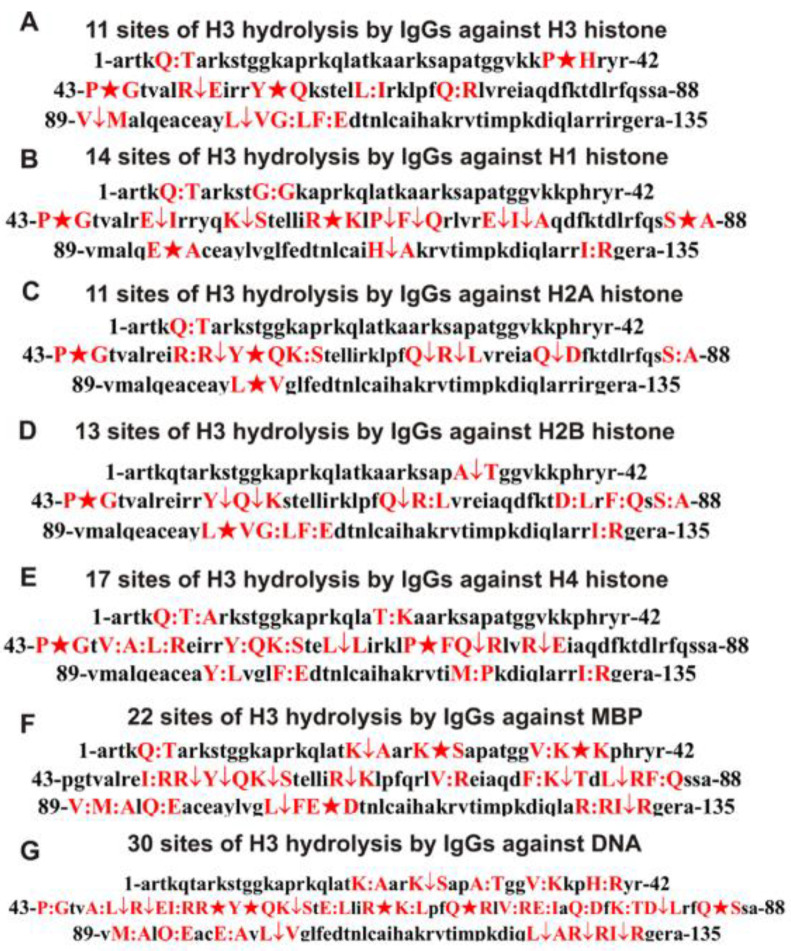
The data on the hydrolysis sites of the H3 histone by IgGs against all five histones (H3 (**A**), H1 (**B**), H2A(**C**), H2B (**D**), and H4 (**E**)), as well as against MBP (**F**) and DNA (**G**). Major cleavage sites are indicated by big stars (★), moderate sites are indicated by arrows (↓), and minor ones are splitted by colons (:) (**A**–**G**).

**Table 1 biomedicines-10-02663-t001:** Sites of H3 histone hydrolysis by IgGs against five histones, MBP, and DNA.

Type of IgGs						
Anti-H3	Anti-H1	Anti-H2A	Anti-H2B	Anti-H4	Anti-MBP	Anti-DNA
11 Sites	14 Sites	11 Sites	13 Sites	17 Sites	22 Sites	29 Sites
Q5-T6 *	Q5-T6	Q5-T6	-	Q5-T6	Q5-T6	-
-	-	-	-	T6-A7	-	-
-	G12-G13	-	-	-	-	-
-	-	-	-	T22-K23	-	-
-	-	-	-	-	** *K23-A24 * ** ***	K23-A24
-	-	-	-	-	K27-S28 *	*K27-S28*
-	-	-	*A31-T32*	-	-	A31-T32
-	-	-	-	-	V35-K36	V35-K36
-	-	-	-	-	K36-K37	-
**P38-H39**	-	-	-	-	-	-
-	-	-	-	-	-	H39-R40
**P43-G44**	P43-G44	**P43-G44**	P43-G44	P43-G44	-	P43-G44
-	-	-	-	V46-A47	-	-
-	-	-	-	A47-L48	-	A47-L48
-	-	-	-	L48-R49	-	** *L48-R49* **
** *R49-E50* **	-	-	-	-	-	** *R49-E50* **
-	** *E50-I51* **	-	-	-	-	-
-	-	R52-R53	-	-	-	-
-	-	** *R53-Y54* **	-	-	** *R53-Y54* **	** *R53-Y54* **
**Y54-Q55**	-	Y54-Q55	** *Y54-Q55* **	Y54-Q55	** *Y54-Q55* **	**Y54-Q55**
-	-	-	** *Q55-K56* **	-	-	-
-	K56-S57	K56-S57	-	K56-S57	** *K56-S57* **	** *K56-S57* **
-	-	-	-	-	-	E59-L60
-	-	-	-	** *L60-L61* **	-	-
L61-I62	-	-	-	-	-	-
-	**R63-K64**	-	-	-	** *R63-K64* **	**R63-K64**
-	-	-	-	-	-	**K64-L65**
-	** *P66-F67* **	-	-	**P66-F67**	-	-
-	** *F67-Q68* **	-	-	-	-	-
Q68-R69	-	** *Q68-R69* **	** *Q68-R69* **	** *Q68-R69* **	-	**Q68-R69**
-	-	*R69-L70*	R69-L70	-	-	-
-	-	-	-	-	V71-R72	V71-R72
-	-	-	-	** *R72-E73* **	-	-
-	** *E73-I74* **	-	-	-	-	E73-I74
-	** *I74-A75* **	-	-	-	-	-
-	-	** *Q76-D77* **	-	-	-	Q76-D77
-	-	-	-	-	F78-K79	-
-	-	-	-	-	** *K79-T80* **	K79-T80
-	-	-	-	-	** *L82-R83* **	-
-	-	-	D81-L82	-	-	** *D81-L82* **
-	-	-	F84-Q85	-	F84-Q85	-
-	-	-	-	-	-	**Q85-S86**
-	**S87-A88**	S87-A88	S87-A88	-	-	-
** *V89-M90* **	-	-	-	-	V89-M90	-
-	-	-	-	-	M90-A91	M90-A91
-	-	-	-	-	Q93-E94	Q93-E94
-	**E94-A95**	-	-	-	-	E94-A95
-	-	-	-	Y99-L100	-	-
** *L100-V101* **	-	L100-V101	L100-V101	-	-	** *L10-V101* **
G102-L103	-	-	G102-L103	-	-	-
-	-	-	-	-	** *L103-F104* **	-
F104-E105	-	-	F104-E105	F104-E105	-	-
-	-	-	-	-	**E105-D106**	-
-	** *H113-F114* **	-	-	-	-	-
-	-	-	-	M120-P121	-	-
-	-	-	-	-	-	** *L126-A127* **
-	-	-	-	-	R128-R129	** *R128-R129* **
-	** *I130-R131* **	-	I130-R131	I130-R131	** *I130-R131* **	-

* Major hydrolysis sites are marked in bold (red), moderate sites are marked in italics (green), minor sites are marked in normal (black), and missing hydrolysis sites are marked with a dash (-).

## Data Availability

Data is contained within the article and Appendix A.

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
