# Peer review of "Multiple Sclerosis: Enzymatic Cross Site-Specific Recognition and Hydrolysis of H3 Histone by IgGs against H3, H1, H2A, H2B, H4 Histones, Myelin Basic Protein, and DNA"

_biomedicines, 2022, doi:10.3390/biomedicines10102663_

Round 1

Reviewer 1 Report

1.  Rephrase this line-Only a few sites of H3 hydrolysis by antibodies against each substrate coincide with those for other ones.

2. Keywords-replace this "Human blood sera antibodies-abzymes" hydrolysis of H3 histone; IgGs against H2B; H1; H2A; H3; H4 histones; enzymatic cross recognition and hydrolysis -with smaller one

3. write the simpler version this definition- Antibodies (Abs) to stable analogs of reaction transition states and natural auto-Abs with catalytic activities are called abzymes (ABZs), and they are successfully described in the literature

4. Rephrase these lines-Some healthy humans and mammals produce antibody-abzymes having low vasoactive intestinal peptide- [16], thyroglobulin- [18], and polysaccharide-hydrolyzing [13-15] activities. However, healthy humans and patients with some pathologies demonstrating insignificant autoimmune reactions usually lack abzymes [1-6]. Nonetheless, germline Abs of some healthy humans could possess amyloid- and superantigen-directed catalytic activities [24,25].

5. MM, rewrite these lines- Then polyclonal IgG preparations were subjected to FPLC gel filtration under drastic conditions (pH 2.6) according to [27-30,32], destroying immune complexes. To analyze the “average” sitespecific splitting of H3 by IgGs against H1-H4 five histones, we have obtained the mixture of equal amounts of fifteen IgG preparations (IgGmix), possessing high activities in the splitting of histones, MBP, and DNA

6. Add more information in figure 1 and 4 legend

7. add cat# for all the reagents

8. The published data is partially overlapping with published data, represent in style-ie.Table 1 findings. I mean it is there in the literaure already. 

9. Discussion must be improved-in second para

10. avoid pronons/prepositions in the begining of sentences.

Author Response

  1. Rephrase this line-Only a few sites of H3 hydrolysis by antibodies against each substrate coincide with those for other ones.

Answer:

I'm sorry, but you propose to replace the phrase “Only a few sites of H3 hydrolysis by IgGs against each of five histones coincide with those for other ones” for “Only a few sites of H3 hydrolysis by antibodies against each substrate coincide with those for other ones”

But then the meaning of the fact that antibodies against five different histones hydrolyze H3 at different sites and only few of them coincide.

Therefore, we changed this phrase to “IgGs against five different histones hydrolyze H3 at different sites and only a few them coincide”.

  1. Keywords-replace this "Human blood sera antibodies-abzymes" hydrolysis of H3 histone; IgGs against H2B; H1; H2A; H3; H4 histones; enzymatic cross recognition and hydrolysis -with smaller one.

Answer: It was done

  1. rewrite the simpler version this definition- Antibodies (Abs) to stable analogs of reaction transition states and natural auto-Abs with catalytic activities are called abzymes (ABZs), and they are successfully described in the literature

Answer: It was done

  1. Rephrase these lines-Some healthy humans and mammals produce antibody-abzymes having low vasoactive intestinal peptide- [16], thyroglobulin- [18], and polysaccharide-hydrolyzing [13-15] activities. However, healthy humans and patients with some pathologies demonstrating insignificant autoimmune reactions usually lack abzymes [1-6]. Nonetheless, germline Abs of some healthy humans could possess amyloid- and superantigen-directed catalytic activities [24,25].

Answer: It was done

  1. MM, rewrite these lines- Then polyclonal IgG preparations were subjected to FPLC gel filtration under drastic conditions (pH 2.6) according to [27-30,32], destroying immune complexes. To analyze the “average” site-specific splitting of H3 by IgGs against H1-H4 five histones, we have obtained the mixture of equal amounts of fifteen IgG preparations (IgGmix), possessing high activities in the splitting of histones, MBP, and DNA.

Answer: It was done

  1. Add more information in figure 1 and 4 legend

Answer: It was done

  1. add cat# for all the reagents

Answer: It was done

  1. The published data is partially overlapping with published data, represent in style-ie.Table 1 findings. I mean it is there in the literature already. 

Answer: Excuse me, but in the literature so far there is only data on the hydrolysis of five histones by total preparations of IgG from patients with multiple sclerosis. In previous work, no separation of antibodies against the five individual histones was performed and the analysis of hydrolysis of the histones was performed using mixture of five histones.

Baranova SV, Mikheeva EV, Buneva VN, Nevinsky GA. Antibodies from the Sera of Multiple Sclerosis Patients Efficiently Hydrolyze Five Histones. Biomolecules. 2019 Nov 15;9(11):741. doi: 10.3390/biom911074.

The data of this work are completely new, including also the hydrolysis of H3 by antibodies with a high affinity for DNA.

  1. Discussion must be improved-in second para

Answer: It's not entirely clear what you mean. Paragraph 2 partially corrected.

We are very grateful for your valuable and helpful comments.

Thank you very much

Prof. Georgy Nevinsky

Reviewer 2 Report

I already reviewed previous articles from the authors. My fundamental scepticism remains : how can one mention specific autoantibodies with catalytic activity in view of the extremely highly diversified proteolytic sites presented by the authors and the immense cross-reactions shown by these autoantibodies. Although the purification procedures seem sound to me, contamination by proteases during and after the purification procedures must be continuously checked using a whole panel of classic protease inhibitors also while subjecting the proteins to the catalytic activity in view of the long incubation times necessary to detect the proteolytic activity. Only referencing article 31 of the authors is not valid enough.

Forty of the 57 references are auto-references. Are the authors the only actors in the field. If it is true, reason the more to be very stringent in the criteria applied to justify the conclusions.

Author Response

  • I already reviewed previous articles from the authors. My fundamental scepticism remains : how can one mention specific autoantibodies with catalytic activity in view of the extremely highly diversified proteolytic sites presented by the authors and the immense cross-reactions shown by these autoantibodies. Although the purification procedures seem sound to me, contamination by proteases during and after the purification procedures must be continuously checked using a whole panel of classic protease inhibitors also while subjecting the proteins to the catalytic activity in view of the long incubation times necessary to detect the proteolytic activity. Only referencing article 31 of the authors is not valid enough.

Answer:

Sorry, but several factors indicate that antibodies do not contain admixtures of classical proteases.

1) All experiments were carried out in conditions of maximum purity

 2) After SDS-PAGE electrophoresis, we detect proteolytic activity only in the zone of the gel corresponding to 150 kDa of IgGs, while all proteases are have mol. masses from 20 to 30 kDa.

Unfortunately, we ran out of preparations to test for the presence of classical protease impurities in each of individual of them. However, as per your remark, an equimolar mixture of IgGs against five histones was used for the analysis. Impurities of canonical proteases using SDS-PAGE were not detected.

3) Each IgG preparation has its own histone hydrolysis sites, and in the case of the presence of impurities, they should be the same for preparations analyzed. Most of the hydrolysis sites do not correspond to those for classical proteases and they are located not along the entire length of the histone, as would be the case with canonical proteases, but in specific clusters.

4) From the text of article

Moreover, the same incarceration can be drawn using the comparison of histone H3 hydrolysis sites with IgGs against MBP, five histones, and DNA. Trypsin hydrolyzes different proteins after the lysine (K) and arginine (R) residues. The H3 sequence contains 13 Lys and 17 Arg residues presenting 30 potential sites for this histone cleavage by trypsin. However, the number of sites of H3 splitting by all IgG preparations used after Lys and Arg varies mainly from 1 (anti-H3, anti-H2B, and anti-H4 IgGs) to 2 (anti-H1 IgGs, anti-H2A IgGs) (Figure 4, Table 1). Only IgGs against MBP cleavage H3 at 10 sites after Lys and Arg disposed within relatively short and specific clusters (Figure 4, Table 1). Chymotrypsin splits proteins after aromatic amino acids (F, Y, and W). There are seven potential such sites for hydrolysis in H3 histone by chymotrypsin. Anti-H2A IgGs did not demonstrate sites of cleavage after F residue (Table 1). Only one site of splitting after F was found for anti-H3 and anti-H1 (Table 1) IgGs, while anti-H2B and anti-H4 IgGs demonstrated two sites of cleavage after F (Table 1). No hydrolysis sites were found after Y in the case of anti-H1, anti-H2B, and anti-MBP IgGs, while one hydrolysis site was revealed for anti-H3, anti-H2A, and anti-H4 Abs. The cleavage sites of H3 by five IgGs occur mainly in clusters containing neutral non-charged and nonaromatic amino acids (Q, P, L, V, G, I, S, and H), or acidic acids - E and D (Table 1, Figure 4). Thus, overall, the sites of specific hydrolysis of H3 by five IgGs against five histones do not correspond to those for trypsin or chymotrypsin. They are not distributed along the entire length of the protein molecule but are located in specific amino acids clusters.

From our point of view, everything has been proven that histone is hydrolyzed by antibodies, and not by impurities.

  • Forty of the 57 references are auto-references. Are the authors the only actors in the field. If it is true, reason the more to be very stringent in the criteria applied to justify the conclusions.

Answer: Unfortunately, there are very few scientific groups that deal with the abzymes of patients with autoimmune diseases. And those, the few who do, are in the first stage of studying abzymes - proof of their existence using total antibody preparations.

We are very grateful for your valuable and helpful comments.

Thank you very much

Prof. Georgy Nevinsky

Round 2

Reviewer 2 Report

The proteolytic activity you mention are compatible with minimal protease amounts, not detectable in SDS page electrophoresis. For me, the only way to be sure no minimal amounts of proteases account for the low enzymatic activity of the IgGs is to perform the proteolysis in presence of a cocktail of protease inhibitors. This control is easy to perform. I do not ask to perform this control for all your IgG preparations. Only a few examples should be sufficient.

Author Response

I'm sorry, but your suggestion for an experiment cannot provide a correct answer.

The fact is that the active centers of abzymes are arranged in the same way as the active centers of classical enzymes. The immune response to molecules that mimic the transient transition states of chemical reactions leads to the formation of the same amino acid residues in antibodies for the substrate recognition sites as in the active centers of canonical enzymes. Secondary - anti-idiotypic antibodies to the active centers of enzymes have a structure similar to the original active centers of enzymes. Both antibodies against transition state analogs and anti-idiotypic antibodies against enzyme active sites have catalytic activity. At the same time, we have shown that individual subfractions of antibodies-abzymes have all types of proteolytic activities that are inherent in canonical proteases - serine-, thiol-proteases and metal-dependent proteases. This has also been shown using monoclonal antibodies abzymes appropriate for patients with systemic lupus erythematosus (see below).

Therefore, treatment of antibody preparations with a mixture of specific protease inhibitors should lead to complete suppression of all protease activities of antibodies. Thus, it is simply not possible to show possible presence of impurities of classical proteases, which have a very high activity, but are not detected after SDS electrophoresis. If the classical proteases are present in the antibody preparations, they would definitely appear after the SDS page at low molecular weight?> but we did no see them.

The suppression of the protease activities of antibodies-abzymes and their light chains using specific protease inhibitors has been shown by us in a number of articles mentioned below.

I quote some data from one of our articles

It was previously shown that several monoclonal light chains corresponding to the phagemid library of recombinant peripheral blood lymphocyte immunoglobulin light chains of patients with systemic lupus erythematosus specifically hydrolyze proteins.

Anna M Timofeeva 1, Valentina N Buneva 1 2, Georgy A Nevinsky  Systemic lupus erythematosus: molecular cloning and analysis of 22 individual recombinant monoclonal kappa light chains specifically hydrolyzing human myelin basic protein J Mol Recognit. 2015 Oct;28(10):614-27. DOI: 10.1002/jmr.2476

Abstract

Antibodies hydrolyzing myelin basic protein (MBP) can play an important role in the pathogenesis of multiple sclerosis (MS) and systemic lupus erythematosus (SLE). An immunoglobulin light chain phagemid library derived from peripheral blood lymphocytes of patients with SLE was used. Small pools of phage particles displaying light chains with different affinities for MBP were isolated by affinity chromatography on MBP-Sepharose, and the fraction eluted with 0.5 M NaCl was used for preparation of individual monoclonal light chains (MLChs, 26-27 kDa). Seventy-two of 440 individual colonies were randomly chosen, expressed in Escherichia coli in a soluble form, and MLChs were purified by metal chelating chromatography. Twenty-two of 72 MLChs have high affinity and efficiently hydrolyze only MBP (not other control proteins) demonstrating various pH optima in a 5.7-9.0 range and different substrate specificity in the hydrolysis of four different MBP oligopeptides. Four MLChs demonstrated serine protease-like and three thiol protease-like activities, while 11 MLChs were metalloproteases. The activity of three MLChs was inhibited by both phenylmethylsulfonyl fluoride (PMSF) and Ethylenediaminetetraacetic acid (EDTA), two other by EDTA and iodoacetamide, and one by PMSF, EDTA, and iodoacetamide. The ratio of relative activity in the presence of Ca(2+), Mg(2+), Mn(2+), Ni(2+), Zn(2+), Cu(2+), and Co(2+) was individual for each of 22 MLCh preparations. It is the first examples of human MLChs, which probably can possess two or even three different proteolytic activities. These observations suggest an extreme diversity of anti-MBP abzymes in SLE patients. The immune systems of individual SLE patients can generate a variety of anti-MBP abzymes, which can attack MBP of myelin-proteolipid sheath of axons and play an important role in MS and SLE pathogenesis.

In addition, a detailed analysis of individual monoclonal antibodies was carried out and it was shown that the structure of their active centers is very close to canonical proteases.

ABSTRACTS

1) It was shown previously that approximately 30% ± 5% of antibodies against myelin basic protein (MBP) and the DNA of patients with systemic lupus erythematosus (SLE) and multiple sclerosis (MS) possess catalytic activities that play an important negative role in the pathogenesis of MS and SLE. An immunoglobulin light chain phagemid library derived from peripheral blood lymphocytes of patients with SLE was used. The small pools of phage particles displaying light chains with different affinity for MBP were isolated by affinity chromatography on MBP-Sepharose, and the fraction eluted with 0.5 M NaCl was used for preparation of individual monoclonal light chains (MLChs, 26-27 kDa). The clones were expressed in E. coli in a soluble form. MLChs were purified by metal chelating chromatography followed by FPLC-gel filtration. The activity of one MLCh (NGTA1-Me-pro) was inhibited only by EDTA, and it efficiently hydrolyzed MBP (but not other proteins) and four different oligopeptides corresponding to four known immunodominant sequences containing cleavage sites of MBP only in the presence of several different metal ions. An unexpected result was obtained: NGTA1-Me-pro demonstrated two pH optima, two optimal concentrations of Me2+ ions, and two Km values for MBP. The protein sequence of NGTA1-Me-pro, having two metalloprotease active centers, has homology with several mammalian metalloproteases. Recently, it was shown that one other MLCh possesses serine-like and metalloprotease activity. The principal possibility of the existence of MLChs with several different active centers is unexpected, but very important for the further understanding of unknown possibilities for immune systems and the biological functions of antibodies.

2) Canonical enzymes usually have only one active site catalyzing some kind of chemical reaction. It was shown previously that in contrast to classical enzymes, preparations of one of the light chains (NGTA2-Me-pro-Tr) showed two optimal pH values, two optimal concentrations of metal ions, and two Km values for MBP. One protease active site of NGTA2-Me-pro-Tr was trypsin like, whereas second one was metal dependent. In this article, a search for protein sequences of NGTA2-Me-pro-Tr responsible for catalytic functions was carried out. We performed, for the first time, analysis of the homology of the protein sequence of NGTA2-Me-pro-Tr with those of several classical Zn2+ - and Ca2+ -dependent, as well as human serine, proteases. The analysis allowed us to identify the protein sequences of NGTA2-Me-pro-Tr responsible for serine-like activity, the binding of MBP, and chelation of metal ions and catalysis directly. The data obtained are summarized using hypothetical models of the structure of the two active centers of a very unusual light chain of antibodies (Abs). The findings obtained may be very important for understanding possible structure of active centers of very unusual light chain of Abs possessing several enzymatic activities.  

In mammalians there are serine proteases, metalloproteases, and DNases. Each of these and many

other enzymes catalyze only one chemical reaction. Very unusual and unpredictable situation was revealed by us in

the case of monoclonal abzymes corresponding to the sera of patients with SLE. The small pools of phage particles

displaying light chains with different affinity for myelin basic protein (MBP) were isolated by affinity chromatography

on MBP-Sepharose. In contrast to canonical enzymes, one of twenty five MLChs demonstrated three different

enzymatic activities; it efficiently hydrolyzed MBP (but not other proteins) and DNA. Other twenty four MLChs

hydrolyzed only MBP. The proteolytic activity of NGTA3-pro-DNase was efficiently inhibited by specific inhibitors of

serine-like (PMSF) and metalloproteases (EDTA). Protease and DNase properties of NGTA3-pro-DNase differ

significantly from those for the corresponding canonical enzymes.

Conclusion: This is the first example of monoclonal antibodies with three different catalytic activities. The

principal possibility of the existence of monoclonal antibodies with several different enzymatic activities is

unexpected but very 

  1. Timofeeva, A. M., Buneva, V. N.,and Nevinsky, G. A. 2016. Systemic lupus erythematosus: molecular cloning and analysis of recombinant monoclonal kappa light chain NGTA1-Me-pro with two metalloprotease active centers. Molecular BioSystems  12:3556-3566.
  2. Timofeeva, A. M., Ivanisenko, N. V., Buneva, V.N., Nevinsky, G. A. 2015. Systemic lupus erythematosus: molecular cloning and analysis of recombinant monoclonal kappa light chain NGTA2-Me-pro-Tr possessing two different activities-trypsin-like and metalloprotease. InternationalImmunology 27:633-645.
  3. Timofeeva, A .M., Buneva, V. N.,and Nevinsky, G. A. SLE: Unusual Recombinant Monoclonal Light Chain NGTA3-Pro-DNase. Possessing Three Different Activities Trypsin-like, Metalloprotease and DNase. Lupus Open Access 2017, 2:127.
  4. Bezuglova, A. V., Buneva, V. N., Nevinsky, G. A. 2011. Systemic lupus erythematosus: monoclonal light chains of immunoglobulins against myelin basic protein possess proteolytic and DNase activities. Russian Journal of Immunology  5:215-227

Results: In mammalians there are serine proteases, metalloproteases, and DNases. Each of these and many

other enzymes catalyze only one chemical reaction. Very unusual and unpredictable situation was revealed by us in

the case of monoclonal abzymes corresponding to the sera of patients with SLE. The small pools of phage particles

displaying light chains with different affinity for myelin basic protein (MBP) were isolated by affinity chromatography

on MBP-Sepharose. In contrast to canonical enzymes, one of twenty five MLChs demonstrated three different

enzymatic activities; it efficiently hydrolyzed MBP (but not other proteins) and DNA. Other twenty four MLChs

hydrolyzed only MBP. The proteolytic activity of NGTA3-pro-DNase was efficiently inhibited by specific inhibitors of

serine-like (PMSF) and metalloproteases (EDTA). Protease and DNase properties of NGTA3-pro-DNase differ

significantly from those for the corresponding canonical enzymes.

And additional References

  1. References: Bezuglova, A. M., Konenkova, L. P., Doronin, B. M., Buneva, V. N., and Nevinsky, G. A. 2011. Affinity and catalytic heterogeneity and metal-dependence of polyclonal myelin basic protein-hydrolyzing IgGs from sera of patients with systemic lupus erythematosus. Journal of Molecular Recognition 24:960-974.
  2. Bezuglova, A. M., Konenkova, L. P., Buneva, V. N., and Nevinsky, G. A. 2012. IgGs containing light chains of the λ-and κ-type and of all subclasses (IgG1-IgG4) from the sera of patients with systemic lupus erythematosus hydrolyze myelin basic protein. International Immunology 24:759-770.
  3. Bezuglova, A. M., Dmitrenok, P. S., Konenkova, L. P., Buneva, V. N., and Nevinsky, G. A. 2012. Multiple sites of the cleavage of 17- and 19-mer encephalytogenic oligopeptides corresponding to human myelin basic protein (MBP) by specific anti-MBP antibodies from patients with systemic lupus erythematosus. 37:69-78.
  4. Timofeeva, A. M., Dmitrenok, P. S., Konenkova, L. P., Buneva, V. N., and Nevinsky, G. A. 2013. Multiple sites of the cleavage of 21- and 25-mer encephalytogenic oligopeptides corresponding to human myelin basic protein (MBP) by specific anti-MBP antibodies from patients with systemic lupus erythematosus. PLoS One 8:e51600.

Sorry, but it is not clear how to show with the help of specific protease inhibitors that there are no possible microimpurities of classical proteases in antibody preparations. From our point of view, all possible controls that could help to do this are presented by us in the article. The only thing that can still be shown is the lack of activity of antibody preparations after their treatment with specific protease inhibitors. But this will not lead to an answer to your question, since the active sites of abzymes have a very close structure to canonical enzymes.

Thanks for the helpful comments

With best regards

Georgy A. Nevinsky

Round 3

Reviewer 2 Report

no comments